# Ameliorative Effect of Structurally Divergent Oleanane Triterpenoid, 3-Epifriedelinol from *Ipomoea batatas* against BPA-Induced Gonadotoxicity by Targeting PARP and NF-κB Signaling in Rats

**DOI:** 10.3390/molecules28010290

**Published:** 2022-12-29

**Authors:** Muhammad Majid, Anam Farhan, Muhammad Waleed Baig, Muhammad Tariq Khan, Yousaf Kamal, Syed Shams ul Hassan, Simona Bungau, Ihsan-ul Haq

**Affiliations:** 1Faculty of Pharmacy, Hamdard University, Islamabad 45550, Pakistan; 2Department of Biology, School of Science and Engineering, Lahore University of Management Sciences, Lahore 54792, Pakistan; 3Department of Pharmacy, Faculty of Biological Sciences, Quaid-i-Azam University, Islamabad 45320, Pakistan; 4Faculty of Pharmacy, Capital University of Science and Technology, Islamabad 44000, Pakistan; 5Shanghai Key Laboratory for Molecular Engineering of Chiral Drugs, School of Pharmacy, Shanghai Jiao Tong University, Shanghai 200240, China; 6Department of Pharmacy, Faculty of Medicine and Pharmacy, University of Oradea, 410028 Oradea, Romania

**Keywords:** *Ipomoea*, aerial part, sweet potato, inflammation, prostate, antiproliferative, apoptosis, pentacyclic triterpenoids

## Abstract

The pentacyclic triterpenoids (PTs) of plant origin are reputed to restrain prostate cancer (PCa) cell proliferation. This study aims to assess 3-epifriedelinol (EFD) isolated from aerial part of Ipomoea batatas against PCa and its potential mechanism, in vitro and in vivo. Molecular docking affirms good binding affinity of the compound with target proteins exhibiting binding energy of −7.9 Kcal/mol with BAX, −8.1 Kcal/mol (BCL-2), −1.9 Kcal/mol (NF-κB) and −8.5 Kcal/mol with P53. In the MTT assay, EFD treatment (3–50 µM) showed a significant (*p* < 0.05 and *p* < 0.01) dose and time dependent drop in the proliferative graph of DU145 and PC3, and an upsurge in apoptotic cell population. EFD displayed substantial IC50 against DU145 (32.32 ± 3.72 µM) and PC3 (35.22 ± 3.47 µM). According to Western blots, EFD administration significantly enhanced the cleavage of caspases and PARP, elevated BAX and P53 and decreased BCL-2 and NF-κB expression, thereby triggering apoptosis in PCa cells. When male Sprague Dawley rats were intoxicated with Bisphenol A (BPA), an apparent increase in prostate mass (0.478 ± 0.08 g) in comparison to control (0.385 ± 0.03 g) indicates prostatitis. Multidose treatment of EFD (10 mg/kg) significantly reduced prostate size (0.404 ± 0.05 g). EFD exhibited substantial curative potential in vivo, as hematological, hormonal and histopathological parameters have been significantly improved. Reduced peroxidation (TBARS), and suppression of inflammatory markers i.e., NO, IL-6 and TNF-α, signposts substantial antiinflammatory potential of the compound. Overall, EFD has shown better binding affinity with target molecules, acceptable ADMET profile, potent antiproliferative and apoptotic nature and significant reduction in inflamed prostate mass of rats. The present study demonstrates acceptable physicochemical and pharmacokinetic properties of the compound with excellent drugable nature, hence EFD in the form of standardized formulation can be developed as primary or adjuvant therapy against PCa and toxins-induced gonadotoxicity.

## 1. Introduction

Numerous therapeutic compounds found in medicinal plants are effective against infectious disorders, cancer, pyrexia, algesia, and inflammation [1,2]. Natural product structures are known to have biological selectivity, significant chemical diversity and structural amity that make them ideal lead structures for developing new drugs [3]. Triterpenoids (TPs), which are isopentenyl pyrophosphate oligomers with significant therapeutic potential, can be found in a variety of plant parts as cyclic (tetra/penta) TPs, phytosterols, glycoside triterpenes conjugates and several other related compounds. TPs can be well classified as oleananes, ursanes and lupanes. The focus to use this class of medications against a wide range of disorders has been drawn by the intriguingly powerful structure of TPs with considerable protein binding affinity. Oleanolic and glycyrrhizic acids, as well as betulinic and gymnemic acids, are commonly used to treat liver and prostatite related problems, corosolic and gymnemic acid for diabetes and associated complications, while asiatic acid to heal wounds. Due to this, the entire family of TPs represents a valuable paragon of unique, bioactive components having diverse disease targets and substantial therapeutic potential [4]. Developing supplements from terpenes-rich plants like *I. batatas* enriched with probiotics like *Streptococcus thermophilus* encourages the development of a healthy intestinal microflora (gerobiotics) with established immune system modulation effect. Consuming such food supplements may also help with the management and treatment of autoimmune and inflammatory disorders [5,6].

One of the most prevalent malignancies in the world with a significant fatality rate is prostate cancer (PCa). In PCa, an enlarged prostate with a resulting decreased urine flow volume and intensity is thought to be the primary cause of sickness and death. With 1.1 million cases, it is the fourth most prevalent kind of cancer on the planet. According to surveys, there would be 170,000 new cases of PCa and 499,000 new deaths by 2030 [7]. The stroma and basal cells’ ability to convert testosterone to dihydrotestosterone (DHT) depends on the intracellular enzyme 5-reductase (5-R). DHT plays a crucial function in prostate expansion and has a 10× stronger affinity for androgen receptors than testosterone [8]. It is difficult to pinpoint the exact epidemiology of sporadic PCa, however it is complex and involves both environmental and genetic components. Exposure to substances like bisphenol A (BPA), alcohol, smoking, carbon tetrachloride (CCl4), thioacetamide, and other xenobiotics is thought to be the main cause of the spread of oxidative stress [9]. BPA (2, 2-bis (4-hydroxylphenyl) propane) is a chemical (xenobiotic) that is overly consumed in the production of polycarbonate plastics, the packaging resin liner of canned foods, dental sealants, dental amalgams, and thermal materials [10]. The body overexpresses reactive oxygen species (ROS), inducible forms of nitric oxide synthase (iNOS), cyclooxygenase-2 (COX-2) and 5-lipoxygenase (5-LOX), when exposed to these stimuli in excess and for an extended period of time. The inflammation and cancer of key organs like the liver, kidney, and testes are closely connected with elevated levels of ROS and iNOS [11,12]. It is generally known that chronic diseases like cancer may result from the infiltration of mediators like NO, interleukins (IL), and prostaglandins (PGs) at the site of inflammation. The primary events, followed by the infiltration of the mediator, are the prevention of apoptosis, increased cell proliferation, suppression of immunity, and heightened cancer cell invasion [13]. In order to tackle high grade resistance and progressing diseases, persistent, focused, and well tolerated therapeutic alternatives are now required. The ability of PTs to regulate abnormalities caused by nuclear factor-B (NF-B), 5-LOX, iNOS, COX-2, and other factors is largely responsible for their anti-inflammatory effects [14].

*Ipomoea batatas* (L.) Lam called “Shakar Kandi” in Pakistan is cultivated worldwide in different countries like USA, China, India, Nigeria, Uganda and Vietnam. Different parts of the plant are locally used for therapeutic purpose as aphrodisiac, anti-microbial, anti-inflammatory, tonic, in burns, mouth and throat ulcers, GIT issues, prostatitis and diabetes [15], 3-epifriedelinol (EFD) also named (3β, 4β, 5β, 8α, 9β, 10α, 13α, 14β) 5, 9, 13-trimethyl-24, 25, 26-trinoroleanan-3-ol, is oleanane triterpenoid isolated for the first time in *I. batatas* aerial part ethyl acetate extract. Chemical formula of EFD is C_30_H_52_O with the molecular weight of 428.72 g/mol. Several studies [16,17,18,19] have claimed significant anticancer potential of EFD against ovarian, breast, leukemia, cervical, and epidermoid carcinomas but no detailed evaluation of EFD against PCa and prostatitis exists. Hence, keeping in mind the excessive use of *I. batatas* in prostatitis and no existing data of EFD use against PCa, the current study has been planned to assess the compound against prostate cancer through a series of in vitro, in silico, and in vivo experiments. This study can help to develop an affordable therapeutic moiety with improved pharmacokinetic and toxicological features to treat PCa and gonadotoxicity from a bio-waste.

## 2. Results

### 2.1. Structure Elucidation of EFD

White needle-shaped crystals of the purified chemical were obtained having an m.p. of 279–283 °C, molecular formula C_30_H_52_O and molecular weight of 428.72 g/mol. The ^1^H NMR spectrum displayed eight distinctive single peaks (δ_H_ 1.031, 3H, H_23_; δ_H_ 0.868, 3H, H_24_; δ_H_ 0.927, 3H, H_25_; δ_H_ 0.935, 3H, H_26_; δ_H_ 0.988, 3H, H_27_; δ_H_ 1.151, 3H, H_28_; δ_H_ 0.945, 3H, H_29_ and δ_H_ 0.976, 3H, H_30_) in the aliphatic up field region. The multiplet peak at 3.71 ppm was of OH hydroxyl residue (H_3_), ratifying that electronegative atom is present, which produced resonance of H_3_ in that region. Remaining protons showed unresolved absorption peaks at 1.2–1.6 ppm. The ^13^C NMR spectrum displayed thirty characteristic carbon peaks giving a clue of triterpenoidal skeleton of the compound. Eight methyls (CH_3_), eleven methylenes (CH_2_), five methines (CH) and six quaternary carbons (C) were detected. As electron withdrawing oxygen atom was present, one carbon (C_3_) resonated at δ_C_ 72.7 confirming the presence of OH group attached. Signal around δ_C_ 49.1 belonged to the methine carbon at C-4. Single crystal XRD analysis of compound crystals was also conducted to confirm the NMR elucidated structure of the compound. Appendix A represents the details regarding crystal’s data and its refinements. All the observed bond angles and bond lengths were found to be normal for the compound. The hydroxyl group present at C_3_ was identified in axial position, while CH group detected at C_4_ was taken as equatorial. The rings (A, B and C) were detected in bowed position as CH groups at C_5_ and C_14_ were repelling each other. Based upon the results of spectroscopic studies and crystallography data, the name (3β, 4β, 5β, 8α, 9β, 10α, 13α, 14β) 5, 9, 13-trimethyl-24, 25, 26-trinoroleanan-3-ol was given to the compound and its trivial name is 3-epifriedelinol (EFD). This compound has already been reported in some plant species, however it was isolated for the first time from *I. batatas*. The ^1^H and ^13^C spectra, XRD proposed 3D structural model are given in the Appendix A and the elucidated structure of EFD is summarized in Figure 1.

### 2.2. Molecular Docking

Utilizing molecular docking, the affinity between the ligands and protein targets was examined. The docking study was conducted using the AutoDock Vina programme via the PyRx user interface. The best docked posture complex and protein’s affinity were evaluated using the E-value (kcal/mol). It offered prediction of the binding constant and free energy for docked ligands. In the current investigation, EFD has demonstrated the most stable complexes, with the lowest binding energies. It exhibited −8.5 kcal/mol E-value with BAX, −8.2 kcal/mol with Bcl_2_, −1.9 kcal/mol with Nf- κB and −8.4 kcal/mol with P_53_. The list of amino acid residues that are involved in binding pocket interactions with EFD and binding energies in comparison to doxorubicin used as standard are tabulated in Table 1. Figure 2 represents 2D and 3D binding interactions of EFD with the amino acid residues at the binding site of BAX (PDB ID: 2K7W) in comparison to doxorubicin used as standard. Additional 2D and 3D images of binding interactions of EFD with target proteins are provided as Appendix A.

### 2.3. Antiproliferative Potential of EFD

Cell viability has been significantly reduced after exposure to EFD for 24, 48, and 72 h. After 48 and 72 h of exposure, it exhibited considerable time-dependent inhibition against DU145, with IC_50_ of 57.24 ± 4.48 µM and 32.32 ± 3.72 µM. While IC_50_ of EFD was recorded as 60.06 ± 4.89 µM after 48 h and 35.22 ± 3.47 µM after 72 h against PC3. Cabazitaxel and Doxorubicin used as positive control exhibited maximum cytotoxicity against DU145 (9.56 ± 1.45 µM and 10.98 ± 2.71 µM, respectively) and PC3 (12.78 ± 2.67 µM and 15.97 ± 2.77 µM, respectively) after maximum exposure time (72 h). Figure 3 represents all the results of the MTT assay.

### 2.4. EFD Induces Apoptosis

Using the Western blot approach, cleavage of PARP and caspase 3 in treated cells, expression analysis of apoptotic (P53 and BAX) and anti-apoptotic (BCL-2 and NF-κB) proteins was evaluated to uncover the potential molecular mechanism of the antiproliferation of cancer cell colonies in the MTT assay (Figure 4A). GAPDH was used as loading control. EFD treatment to PCa cells (DU145) was done for 48 h at two different concentrations (5 µM and 10 µM). Protein (40–60 µg) was separated from the protein lysate and probed with particular monoclonal antibodies. Cleavage of PARP via caspase 3 is believed to be a vital sign of apoptosis. The expression of P53 and BAX is up regulated whereas NF-κB and BCL-2 are down regulated, suggesting that EFD causes apoptosis of various degrees. Figure 4B displays the fold change in protein expression after a 48 h treatment.

### 2.5. Size, Weight and ROW of Gonads

On day 21 of the experiment, measurements were made on the removed gonads. The data (Table 2) show that the prostate size of BPA-affected rats may have increased (*p* < 0.05), when compared to the control (0.385 ± 0.13 g) exhibiting ROW (0.274 ± 0.16). As the BPA group lost the most weight in the testes (1.68 ± 0.29 g) and epididymis (0.334 ± 0.17 g), it is clear that BPA causes gonadal toxicity. The BPA + EFD (10 mg/kg) and BPA + EFD (05 mg/kg) groups have provided maximum protection, and both showed a substantial (*p* < 0.05) decrease in prostate weight (0.404 ± 0.05 and 0.429 ± 0.04 g, respectively). EFD’s safety in multidose therapy was validated by the test group fed with EFD (10 mg/kg) only, which demonstrated no noticeable increase in prostate size (0.388 ± 0.06 g).

### 2.6. Effect on Hematology and Histology

Table 3 lists the complete blood profiles of the BPA-toxicated rats given EFD (10 mg/kg and 5 mg/kg) dosages. High degree toxicity was caused by the BPA intoxication’s inducing alteration to numerous haematological parameters. When compared to control, BPA-treated rats showed a significant decline in the Hb level, platelets and RBC’s along with elevated WBC count and ESR.

Contrary to that, EFD (10 mg/kg) test groups did not exhibit any detectable alterations in hematologic parameters when compared to the control group, supporting its biosafety and application in in vivo systems. In comparison to control, RBC count (5.51 ± 0.12 and 5.12 ± 0.31 × 10^6^/µL), WBC’s (4.32 ± 0.14 and 4.85 ± 0.17 × 10^3^/µL), platelet count (451.2 ± 1.71 and 405.1 ± 1.91 × 10^3^/µL), Hb (11.26 ± 0.71 and 10.12 ± 0.25 g/dL), and ESR (4.31 ± 0.072 and 5.12 ± 0.101 mm/h) were significantly restored in the test groups. The haematological profile of treated rats has significantly changed as a result of BPA intoxication. Histological studies confirm EFD’s gonadoprotective ability and confirm its anti-prostate prowess. Slides demonstrate that EFD is preemptive against BPA-induced toxicity at both its high (10 mg/kg) and low (5 mg/kg) doses (Figure 5). Spermatocytes, spermatids, spermatogonia, cells of Sertoli and Leydig, normal architecture of seminiferous tubules, normal developmental stages, and concentration of sperm in the seminiferous tubules were all present in the control group’s testes. BPA significantly damaged and abraded seminiferous tubules with low cellular density, while test groups exhibited considerable protection in terms of the morphology of the seminiferous tubules and high density of germ cells.

### 2.7. Effect of EFD on Gonadal Hormones

Sex hormones, i.e., testosterone, LH, estradiol and FSH, were measured in EFD (10 and 5 mg/kg)-treated rats and compared to control and vehicle showed no significant (*p* < 0.05) changes. Hence, fortifying the non-toxic attitude of EFD towards testicular mass and sex hormones. Contrarily, test groups BPA + EFD (05 and 10 mg/kg) demonstrated substantial protection towards BPA intoxication as testosterone (4.02 ± 0.09 and 3.81 ± 0.16 ng/mL), FSH (10.12 ± 0.27 and 8.62 ± 0.36 mIU/mL, LH (2.90 ± 0.13 and 2.49 ± 0.12 mIU/mL), and estradiol (21.07 ± 1.17 and 22.82 ± 0.88 pg/mL) concentrations were measured. Estradiol levels in rats exposed to BPA were elevated (26.19 ± 2.16 pg/mL), clearly showing gonadotoxicity, which eventually causes prostatitis (Table 4). EFD has effectively retained physiological serum estradiol concentrations in a dose-dependent manner, strongly indicating its potential to prevent prostate cancer.

### 2.8. Effect of EFD on Endogenous Enzymes and Inflammatory Markers

Table 5 enlists relative concentrations of antioxidant enzymes in testicular homogenates of rats treated with EFD in comparison to controls. The BPA-intoxicated group showed paramount alterations in the level of enzymes, which is indicative of high degree gonadotoxicity. While restoration of the endogenous antioxidant enzymes in the test groups BPA + EFD (10 mg/kg) and BPA + EFD (05 mg/kg) was significantly (*p* < 0.05) improved. Serum levels of pro-inflammatory markers in BPA-intoxicated rats has radically increased, indicating high degree perturbations. The substantial gonadoprotective ability of EFD is attributed to its significant antiinflammatory potential that can be observed in the test groups receiving EFD (10 and 05 mg/kg) along with BPA (Figure 6).

### 2.9. Pharmacokinetic and ADMET Properties

The physicochemical properties of EFD are mentioned in Table 6. The EFD physicochemical features are categorized into six main groups with appropriate ranges for oral bioavailability (Figure 7A). The molecule strictly follows the schemes; lipophilicity (LIPO) 0.7 < (Log *P_o/w_*) XLOGP3 < +5.0, size (SIZE) 150 g/mol < MV <500g/mol, polarity (POLAR) 20Å2 < TPSA < 130 Å^2^, Insolubility (INSOLU) 0 < Log S (ESOL) < 6, Insaturation (INSATU) 0.25 < Fraction Csp3 < 1 and flexibility (FLEX) 0 < No. of rotatable bonds < 9. The Topological polar surface area (TPSA) values of EFD ranged from 20.2 Å^2^ to 130.0 Å^2^, indicating that it has excellent transport qualities in vivo. Figure 7a depicts the oral bioavailability radar chart graphed on the bases of six physicochemical parameters highlighted in Table 6. The compound EFD produced scores within these criteria, demonstrating that EFD has a favorable physiochemical profile, which is one of the parameters required in medicines or clinical trials.

Gastrointestinal absorption (HIA) and CNS absorption are key variables for every biomolecule before it enters pharmaceutical or clinical trials for drug development. In order to assess the effects on CNS, the blood–brain barrier (BBB) penetration is crucial. This is because substances that affect the CNS must pass through BBB, whereas substances that do not affect the CNS should not interact to avoid adverse effects. The compound EFD exhibited low HIA with no BBB penetrability.

The BOILED-EGG graph, which forecasts HIA and BBB permeation of EFD, is shown in Figure 7B. The yellow area (yolk) is the BBB permeation zone while the white area represents GI absorption zone (HIA). If a compound falls in the grey zone, it suggests that it is neither absorbed in GI nor can it penetrate BBB to reach CNS. As EFD is not a P-gp (P-glycoprotein) substrate, it is not responsive to the P-gp efflux mechanism, which many cancer cell lines use for the development of drug resistance. Furthermore, the compound EFD has a lower skin permeation Log Kp (Table 7); the lower the Kp, the lesser skin permeant is the molecule.

Additionally, this program prophesies the five main cytochrome (CYP) isoforms. These enzymes are essential for the excretion of medicines, and these isoforms metabolize roughly 75% of the commercially available medications. Any of these isoforms, if inhibited, result in serious pharmacokinetics-based medication interactions. According to the data in Table 7, the compound EFD does not block any cytochrome isoform and is rapidly metabolized, so it does not have the potential to cause any drug–drug interactions with the cytochromes that were selected. Drug clearance, which is the sum of hepatic and renal clearances in case of excretion, is a key parameter in determining dosing rates to achieve steady-state concentrations. However, the EFD’s clearance value was insufficient. Substrates of Organic Cation Transporter 2 (OCT2) may affect the potential for negative interactions with OCT2 inhibitors. Swiss ADMET tool predicted that EFD is not an OCT2 substrate. AMES toxicity test that was devised by Bruce Ames, is used to assess the carcinogenic ability of a chemical substances by their mutating effect on bacterial strain *Salmonella typhimurium.* Human ether-a-go-go-related gene (hERG) toxicity test is indicative of cardiotoxicity. Chemicals that bind to potassium channel in heart may produce long QT syndrome (LQTS) resulting ventricular arrhythmias and sudden death. Similarly, the fathead minnow toxicity test is also used to assess the toxicity of chemicals by exposing larvae of fathead minnows (*Pimephales promelas*) to different concentrations. As tabulated in Table 7, EFD exhibited no AMES toxicity, no hepatotoxicity, very low to no minnow toxicity, no skin sensitization and no hERG I and II inhibition, which sanctions its use as lead compound.

## 3. Material and Methods

### 3.1. Chemicals and Reagents

Purified EFD was thoroughly dissolved in dimethyl sulfoxide (DMSO) and stock solution was stored at −20 °C. Stock solutions of the positive controls (Doxorubicin and Cabazitaxel obtained from Sigma, Ronkonkoma, NY, USA) were also prepared in DMSO. Cell Signaling Technology (Beverly, MA, USA) supplied primary antibodies i.e., Bcl2, P53, Nf-kB, PARP and cleaved Caspase 3 while anti-mouse and anti-rabbit secondary antibodies were procured from GE healthcare (Pittsburgh, PA, USA). PCa cell lines i.e., PC3 [CRL-1435] and DU-145[HTB-81] were ordered from ATTC; Manassas, VA, USA. DMSO, BPA, TCA, and TSB used in the experiments were purchased from Sigma while tween 20 from Merck (Rahway, NJ, USA). Medium 99, FBS, RPMI-1640, DMEM, PBS, and MTT powder were procured from Merck Millipore (Burlington, MA, USA). All sized silica gel plates and chromatography columns were ordered from Merck, Darmstadt, Germany.

### 3.2. Preparation of Samples

The whole isolation scheme of EFD from *I. batatas* is attached in Appendix A. Purified compound in the form of crystalline needles was eluted with n-hexane and EA (1:0–0:1) using flash column. Purity of the compound was detected using LC-PDA-ELSD method from a single peak acquired during analysis. After characterization via X-ray crystallography, 1D and 2D NMR, FTIR and Mass spectroscopy, EFD was completely dissolved in 10% DMSO and stored as stock solution for further use.

### 3.3. Molecular Docking

The Autodock Tools Program was utilized in order to complete the preparation of the protein of interest. 3D structures of the target proteins, i.e., Bcl2 (Human gamma herpes virus 8, PDB-ID 1K3K), BAX (Homo sapian, PDB-ID 2K7W), P53 (Homo sapiens, PDB-ID 1AIE) and NF-κB (Musmusculus PDB-ID 1NFK) were retrieved from the protein data bank (www.rcsb.org accessed on 20 February 2022) with PDB IDs of 1M9K and 1O86, respectively. After energy minimization and the addition of Gasteiger charges, proteins structures were saved in PDBqt format. Ramachandran plots and hydrophobicity data were generated using Discovery Studio 4.1 Client (2012). VADAR 1.8 was used to evaluate the protein structure and the statistical percentage values of helices, sheets, coils, and turns [20,21]. Detailed ligands molecular docking and structure analysis of the target proteins is attached in the Appendix A while Ramachandran plots are in in Appendix A.

### 3.4. Cytotoxicity against Prostate Cancer Cell Line

Following pre-established procedure, the MTT assay was performed on the PCa cell lines DU145 and PC3 to gauge the cytotoxic potential of EFD [22]. Briefly, DU145 and PC3 cells were treated for 24, 48, and 72 h with various concentrations of EFD and the % age of viable cells was computed by setting the cell viability of untreated as 100%. The experiment was repeated three time and IC_50_ was determined for the treatment periods, i.e., 24, 48 and 72 h.

### 3.5. Western Blotting

The previously reported methodology was followed to extract proteins and to analyze via the Western blot technique [23]. Cells (DU145) with density 1 × 10^6^/flask were treated for 24 h with EFD (10 μM and 20 μM). After treatment, the media was aspirated, cells were rinsed in cold PBS with a pH of 7.4, trypsinized, and pelleted in 15 mL falcon tubes. The pellet was then treated with cold lysing buffer. A fraction of the sample buffer was carefully mixed to 40–60 μg of protein which was then then incubated at 95 °C for 10 min for denaturation. After samples had cooled to room temperature, they were briefly centrifuged. 50 cl of denatured sample was mixed with 5 μL of marker, loaded in each well of loading gel and then fixed using mini-Protean^®^ Tetra system vertical electrophoresis tank. Running buffer was added to the tank, and proteins were resolved using Power Pac Firmware version 1.07 at 3 A and 100 V for 1.5 h. Following separation, proteins were transferred using the Trans-Blot TurboTM transfer device to a 0.2 m nitrocellulose membrane at 2.5 A for 10 min. Blots were washed for 5 min with wash buffer (1×) after transfer and blocked for half an hour using blocking buffer. After blocking, blots were probed with suitable primary antibody (3 μL/3 mL blocking buffer) for overnight at 4 °C. The blots were then washed for 5 min with wash buffer (1×), probed for 2 h at room temperature with a particular secondary antibody (mouse/rabbit IgG), and developed for 5 min with ECLTM Prime Western blotting detection reagent. Chemiluminescence autoradiography was used to find protein bands using the ChemDocTM MP imaging system (Bio-Rad, Hercules, CA, USA).

### 3.6. Animals

Male Sprague-Dawley rats, having weight (180–200 g) and age of 9–10 weeks were purchased from the National Institute of Health (NIH). All the experimental animals were familiarized to the environment for one week prior to the induction of BPA-induced gonadotoxicity. The animal housing was kept at a constant temperature of 23 ± 0.5 °C and relative humidity of 50%–60%, with a 12 h light–dark cycle and full access to food and water. The “Principles of Lab Animal Care” from NIH were strictly followed in all animal experimental models. The investigation was designed to possibly cause the least amount of distress, discomfort, and pain to the test animals. The Quaid-i-Azam University’s ethics committee established criteria for the care and use of animals in research, and these were rigorously adhered to the letter (QAU-PHM-017/2016).

### 3.7. Experimental Design

A thorough examination against BPA-induced gonadotoxicity was undertaken in male Sprague-Dawley rats (*n* = 7) in order to evaluate the protective ability of EFD. The experiment lasted a total of 21 days and was conducted as previously described by [24] using the published methodology of [25]. This study comprised the experimental groups as the control group, vehicle, BPA-induced only, EFD (10 mg/kg) receiving only, BPA + EFD (10 mg/kg) receiving group and BPA + EFD (5 mg/kg) receiving group. Throughout the entire experiment, the clinical scoring, survival, and body weight changes were tracked. The rats were put to death by cervical dislocation after anesthesia with xylazine and ketamine injections (16 mg + 60 mg, i.p.) to prevent discomfort. The institution’s ethical committee monitored the euthanasia procedure. The animal’s movement, heart rate, respiration, and eye reflex were all observed before disposal.

### 3.8. The Gonads’ Size, Weight, and Relative Organ Weight (ROW)

After euthanizing rats on the final day, as described above, gonads were removed, freed of excess connective tissue, weighed, and sized. ROW was determined using formula
$$\mathrm{ROW}\;=\left(\frac{\mathrm{AOW}}{\mathrm{BW}}\right)\times 100$$
where AOW stands for absolute organ weight in gram while BW is body weight in gram.

### 3.9. Hematological Parameters

For haematological, biochemical, and serological analyses, blood samples were taken via the abdominal aorta while rats were under anesthesia. Blood samples were centrifuged at 6000 rpm for 15 min at 4 °C to separate the serum, which was then either tested or kept at 20 °C. Red blood cells (RBC), platelets, and white blood cells (WBC) were counted using a Neubauer hemocytometer. Hemoglobin (Hb) content was calculated using the Sahli’s haemoglobin meter. In order to measure the erythrocyte sedimentation rate (ESR), the modified Westergren method was used [15].

### 3.10. Hormonal and Biochemical Assessment

The amounts of testosterone, follicle-stimulating hormones (FSH), luteinizing-hormone (LH), and that of estradiol were measured using a standard technique as earlier described [24] to investigate how BPA intoxication affects sex hormones levels while EFD restrains their serum concentrations. The Astra Biotech kit (Berlin, Germany) was used to determine serum testosterone levels at a sensitivity of 0.2 nmol/L–50 nmol/L. Using an ELISA reader, an immunological enzymatic approach was used to test these hormones.

### 3.11. Endogenous Enzymes and Pro-Inflammatory Markers

Using predefined protocols [26,27], the concentrations of Catalase (CAT), peroxidase (POD), superoxide dismutase (SOD), glutathione (GSH), nitric oxide (NO) and lipid peroxidation (TBARS) were measured. By measuring the rate of H_2_O_2_ hydrolysis at 240 nm per minute, the activities of CAT, POD, and SOD were assessed. One unit of catalase activity was defined as an absorbance change of 0.01 units per minute and expressed as unit per milligram protein (U/mg protein). GSH was calculated at 405 nm as serum oxidation with DTNB (μM/mg protein). Similarly, NO was gauged as a reduction of Griess reagent at 540 nm. NO quantification (μM/mg protein) in serum was performed using sodium nitrate curve. The TBARS was quantified as nM TBARS/min/mg at 535 nm using thiobarbituric acid reagent. The molar extinction coefficient (1.56 × 10^5^/M/cm) was used to measure lipid peroxidation at 37 °C. Proinflammatory cytokines viz interlukin-6 ( IL-6 Cat# BMS603-2) and tumor necrosis factor-alpha (TNF-∝, Cat# BMS607-3) were assessed in serum using ELISA kits by following the manufacturer’s instructions (ThermoFisher Scientific, Waltham, MA, USA).

### 3.12. ADMET Predictions

ADMET (Absorption, distribution, metabolism, excretion and toxicity) are critical assessment methods for any molecule before it is approved as a therapeutic candidate. The ADMET attributes of the EFD were obtained using the online web application swiss ADME (http://www.swissadme.ch/index.php, accessed on 20 February 2022) [28], while the pharmacokinetic scores were projected using the online web application pkCSM (http://biosig.unimelb.edu.au/pkcsm/prediction, accessed on 20 February 2022).

### 3.13. Statistical Analysis

Entire data of the study is reported as Mean ± SD. By using Statistix 8.1, a one-way analysis of variance was conducted to ascertain the variation across groups. Different graphs were plotted using Graph Pad Prim 8.1. Significant differences between groups were determined using Tukey’s multiple comparison at *p* < 0.05 and *p* < 0.01.

## 4. Discussion

The following are the study’s key findings: Characterization of 3-epifriedelinol, molecular docking-based assessment of EFD’s affinity for pro- and anti-apoptotic proteins, activation of apoptosis at very low concentrations, reduction of prostate cancer cell proliferation, and gonadoprotection.

Due to their lower risk of adverse effects, and superior therapeutic performance phytoconstituents have shown great promise in the research and development of anticancer treatments. In recent years, numerous powerful triterpenoids derived from plants have been identified, exhibiting significant potential as chemopreventive and chemotherapeutic drugs by downregulating key signaling molecules, inflammatory mediators, tumor cell proliferation, and angiogenesis in a variety of in vitro and in vivo cancer models.

Epifriedelinol was isolated as needle shaped white crystals with m.p. 279–283 °C, molecular formula C_30_H_52_O and molecular weight of 428.72. Eight distinctive singlet peaks of the aliphatic up field region detected via ^1^H NMR and proton under hydroxyl residue (H-3) showed a multiplet peak at 3.71 ppm, confirming the presence of the electronegative atom, which caused the resonance of H-3 at this area. The ^13^C NMR spectrum showed thirty carbon peaks giving an idea of triterpenoidal nature of EFD. It revealed the presence of eight methyls, eleven methylenes, five methines and six quaternary carbons. Due to the existence of electron with drawing oxygen atom, one carbon resonated at δC 72.7 confirming the presence of hydroxyl group. The ability to substitute at C-28 to generate an acid or ester of EFD and the presence of an OH group at C-3 both significantly increase the cytotoxic action of these compounds [29]. In the current study, EFD primarily demonstrated a significant reduction in the ability of prostate cancer cells to proliferate and induction of apoptosis by changing the expression of pro and anti-apoptotic proteins. Apoptosis is a difficult process to fully comprehend, although it can be separated into extrinsic (FAS/FasL ligand) and intrinsic (mitochondrial) routes in general. A protein complex called NF-κB regulates the synthesis of cytokines, the transcription of specific genes, and cell viability. In particular, TRAF1/TRAF2 are regulated by NF-κB, which also inhibits the caspase family of enzymes, which is implicated in multiple apoptotic pathways. Therefore, downregulating NF-κB will cause caspases to be activated and eventually upregulate apoptosis [30]. P53 suppresses the tumor through activating the expression of yet another apoptotic protein, BAX. Therefore, overexpression of P53 and BAX intensifies the cell’s planned death. BCL-2 is an antiapoptotic protein that prevents mitochondrial apoptosis by preventing the release of cytochrome c, which prevents caspase 3 from being activated [31]. Triterpenoids having a freidelane structure have reportedly demonstrated strong antiproliferative action against the tumour cell lines A549, HEPG2, MDA-MB-231, myelomas, MCF-7, and B16F10 [32,33,34,35]. Previously EFD has been found to induce prompt apoptosis in human cervical carcinoma cells [36]. Significant % inhibition of cancer rat glioma (C6) and THP-1 cells by EFD isolated from *A. tataricus* has been reported [37,38]. Extensive study has reported significant anticancer potential of isodeoxyelephantopin and epifriedelinol isolated from *E. scaber* against lung cancer (A549) and breast cancer (T47D) cells [39]. Earlier Rengarajan, et al. [40] reported that D-pinitol instigate apoptosis in breast cancer (MCF-7) cells by expressing BAX and P53 while down regulating BCL-2 and NF-kB level. Numerous studies have shown that ursolic acid can cause apoptosis in a variety of cancer cells by upregulating the pro-apoptotic proteins BAX and P53 and downregulating the anti-apoptotic proteins Bcl-2 and NF-kB [41,42,43]. Drug likeness of the molecule and acceptable pharmacokinetics endorse the development of EFD into formulation against prostatitis, as molecules have potential to induce apoptosis in cancer cells.

Continuous confrontation with BPA, like xenobiotics, results in oxidative stress, which induces genitourinary alterations, increased prostate bulk, and epididymal shrinkage [44]. It is obvious in the current study that prostate mass (0.478 ± 0.28 g) of BPA intoxicated rats has significantly (*p* < 0.05) increased, while epididymal mass (0.334 ± 0.17 g) is reduced. Currently, male Sprague Dawley rats’ prostate mass has increased (*p* < 0.05) after receiving multiple doses of BPA, but their epididymal weight has decreased (*p* < 0.05). Triterpenoids with oleanane, ursane, and lupane skeletons have significant antioxidant nature and a greater tendency to inhibit ROS and iNOS-induced pathological conditions in the body [45]. As OH and COOH have an admirable H+ donating property, their presence at positions C-3 and C-29 causes PTs to become more antioxidant molecules [46]. EFD’s protective potential was shown when it drastically reduced prostate growth and increased epididymal mass in comparison to control and BPA-impaired rats. Previously, Ayyanar and Subash-Babu [47] discussed in detail that PTs in *S. cumini* have displayed significant antioxidant and gonadoprotective effects. Likewise, Olasantan et al. [48] reported gonadoprotective aptitude of *A. floribunda* PTs.

Hematology is considered as a key indicator of the presence and severity of any type of inflammation. Incessant oxidative stress affects vital organs like the testicles by causing inflammation that can be noticed by hematopathological exams. It is well documented in the literature that greater levels of circulatory NO, a raised ESR, a larger WBC count, a lower platelet count, and uremic toxicity are all related [49]. Intoxicated rats with BPA showed all of these abnormalities in their haem parameters, which suggests oxidative stress. Hematological values of the test groups were significantly preserved by co-treatment with EFD at both higher and lower doses. This is the first account of the biological efficacy of EFD in reversing oxidative stress-related haematological markers. Gonadotropin levels in serum are altered by ongoing oxidative stress caused by ROS. In the current study, BPA toxicity is to blame for the long-term production of free radicals that disrupted gonadotropin levels, resulting in gonadotoxicity and enlarged prostate mass. Significant ROS-induced aberrations were not observed in test groups receiving EFD (10 and 5 mg/kg), and significant gonadotropic hormone concentrations were quantified. Ayyanar and Subash-Babu (38) already went to great length in the promising antioxidant and gonadoprotective properties of terpinoids in *S. cumini*. BPA-induced testicular damage are related with a reduction in the amount of endogenous antioxidant enzymes and an increase in nitrite production. Elevated nitrite concentrations in BPA-intoxicated rats indicate vascular endothelial injury or neutrophil activation in damaged testicular tissue, both of which resulted in NO generation. As compared to control and vehicle groups, the EFD (5 and 10 mg/kg) treated groups showed no discernible variations in CAT, SOD, or POD. Oxidative stress is the end result of prolonged bodily exposure to endocrine disrupting chemicals (EDCs) such as BPA. Subsequently hypomethylation, genetic variations and testicular disturbance occur ensuing immature and demorphed spermatogenesis [22,50,51]. Marked antioxidant, antiproliferative and anti-inflammatory aptitude endorses EFD’s anticancer potential. Developing supplements from EFD enriched with probiotics that encourages the development of a healthy gut may help with the management and treatment of EDC’s induced inflammatory disorders.

Before an isolated compound is transformed into a drug, the ADMET profiling is essential for all types of biomolecules. It is evident from ADMET data that EFD exhibits moderate absorption and distribution, lesser GI solubility and no BBB permeability. Hence EFD has fewer to no chances of toxicity and adverse effect in GIT and CNS. As the infiltration of inactive biomolecules towards CNS should be avoided to prevent deleterious effects on the CNS. Moreover, EFD has demonstrated that it is not a P-gp (P-glycoprotein) substrate; as a result, EFD is resistant to the P-gp efflux mechanism, which is used as a drug resistance process by many cancer cell lines. CYP enzyme family plays a significant role in drug metabolism and excretion, metabolizing about 75% of marketed drugs. If any of these isoforms get inhibited, severe pharmaco-kinetics-based medication interactions may occur [21]. EFD showed no CYP enzymes inhibition, suggesting EFD cannot cause drug interactions for those CYP enzyme-targeted medications. Carcinogenic potential, hepatotoxicity, cardiotoxicity and skin sensitization are the areas of concern for any lead subjected to drug development. Data indicate that EFD has not revealed any such toxicity. Current study strongly imply that EFD has the potential to inhibit the multifaceted development of PCa and EDC-induced gonadotoxicity. As a result, a thorough evaluation of standardized EFD pharmaceutical preparations as primary or supplementary therapy in males with PCa and testicular injuries is required.

## 5. Conclusions

The current investigation fortifies the ethnomedicinal usage of *I. batatas* for gonadotoxicity and shows that 3-epifriedelinol, a powerful terpene compound, isolated from the aerial part, have potential to treat prostate cancer. EFD has shown significant antiproliferative aptitude and can cause PCa cells to undergo apoptosis. In the animal model, the compound has exhibited significant tumor regression, restored haematological parameters, reinstated serum hormones, and suppressed inflammatory markers, which fortifies its anticancer knacks. Further research into the molecule may be beneficial in establishing it as a viable therapeutic substitute for the management and treatment of genital abrasions and PCa.

## Figures and Tables

**Figure 1 molecules-28-00290-f001:**
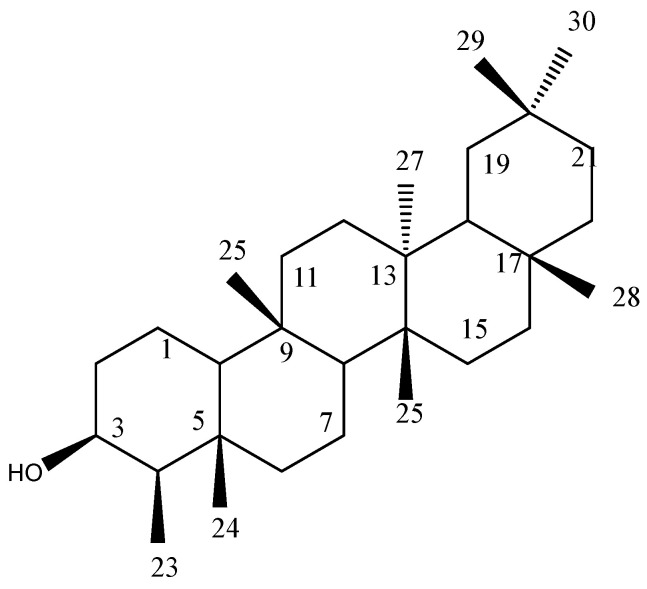
Chemical structure of EFD.

**Figure 2 molecules-28-00290-f002:**
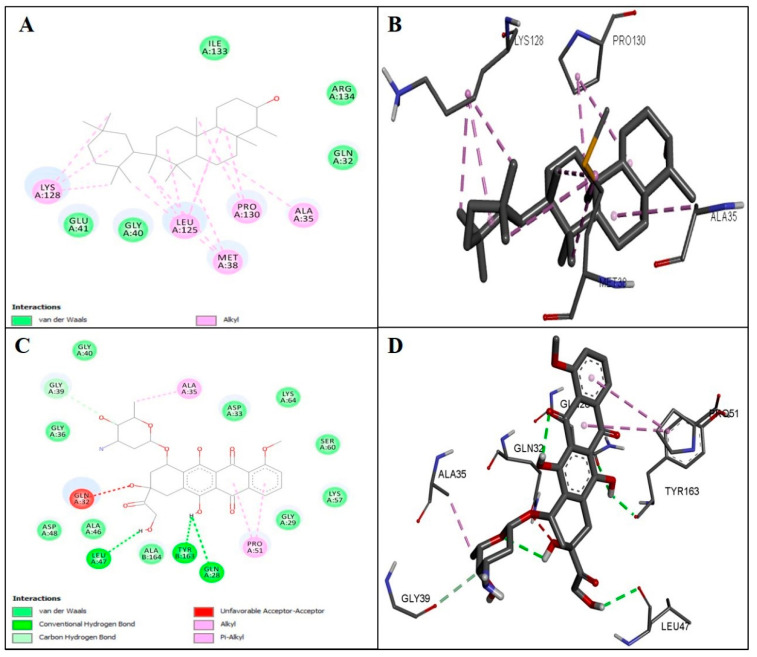
Representation of docked ligands to BAX (PDB-ID 2K7W). (**A**) 2D (**B**) 3D images of EFD presenting binding with BAX. (**C**) 2D (**D**) 3D images of Doxorubicin presenting binding with BAX. Van der Waals, H-bonding, unfavorable acceptor-acceptor, Pi-anion, Pi-sigma, alkyl and Pi-alkyl interactions are involved in ligands binding in protein pockets.

**Figure 3 molecules-28-00290-f003:**
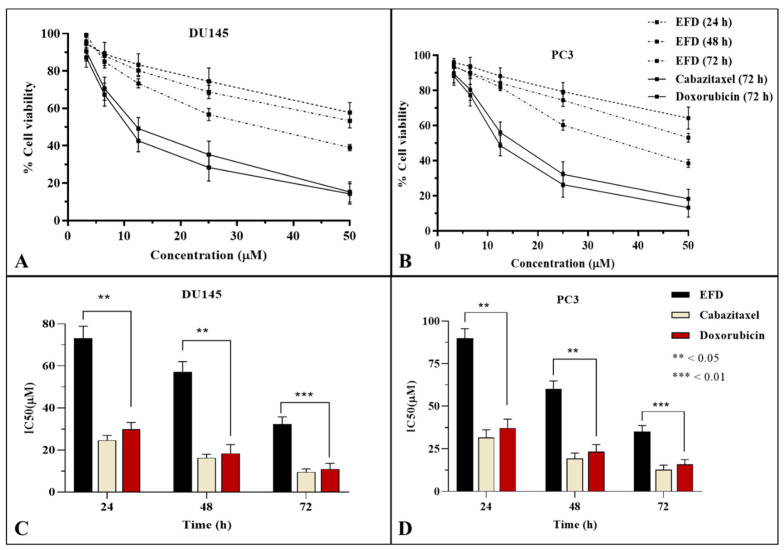
Effect of EFD on viability of prostate cancer cell and IC_50_ values. Note: MTT assay was used to determine the viability of cancer cells (**A**) 24, 48 and 72 h treatment of DU145 cells and (**B**) 24, 48 and 72 h treatment of PC3 cells, (**C**) IC_50_ values of EFD against DU145 (**D**) IC_50_ values of EFD against PC3. Data is mean ± SD of % cell viability (*n* = 3) at ** *p* < 0.05 and *** *p* < 0.01.

**Figure 4 molecules-28-00290-f004:**
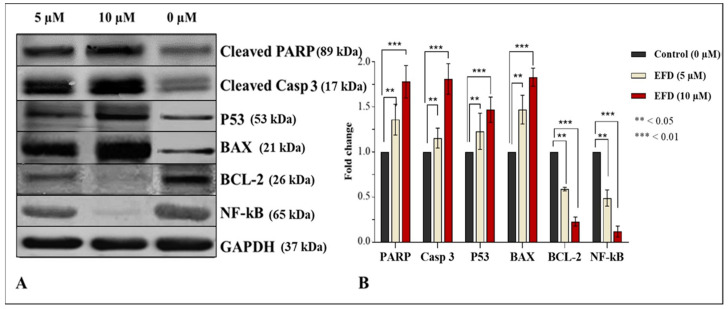
Western blot analysis of proteins associated with EFD induced apoptosis. Note: (**A**) The data demonstrate enhanced caspase 3 cleavage and downstream PARP, upregulation of pro-apoptotic P53 and BAX while downregulation of antiapoptotic NF-κB and BCL2. The loading control used was GAPDH. (**B**) Fold change comparative to control in cleaved PARP, Casp 3, P-53, BAX, BCL-2 and NF-κB expression after 48 h treatment with EFD.

**Figure 5 molecules-28-00290-f005:**
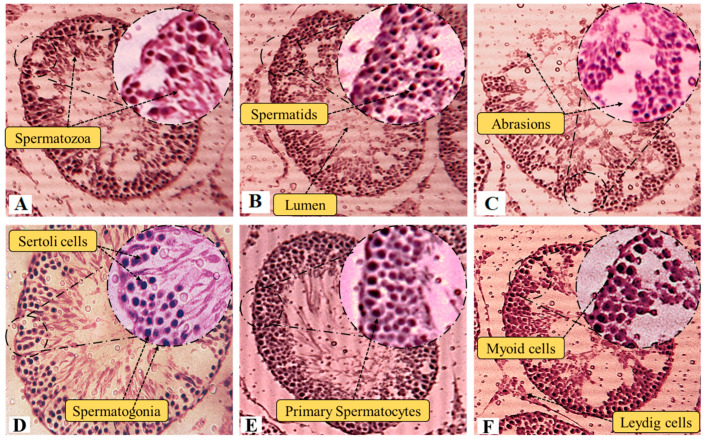
Histological assessment of the protective efficacy of EFD on rat testes. Note: 40× Hematoxylin-eosin stain. (**A**) Control, (**B**) Vehicle, (**C**) BPA (**D**) EFD (10 mg/kg) (**E**) BPA + EFD (10 mg/kg) (**F**) and BPA + EFD (5 mg/kg).

**Figure 6 molecules-28-00290-f006:**
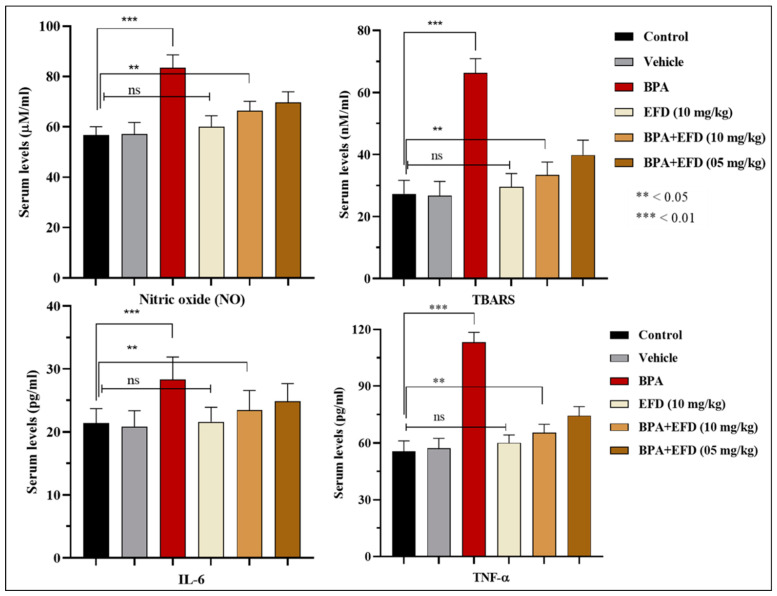
Fate of pro-inflammatory markers in experimental animals after co-treatment with EFD (05 and 10 mg/kg). (NO) Nitric oxide, (TBARS) Thiobarbituric acid reducing substances (Lipid peroxidation assay), (IL-6) Interleukin-6, (TNF-α) Tumor necrosis factor alpha. Data is presented as mean ± SD (*n* = 7). Differences in serum levels were considered significant at the level of ** *p* < 0.05 and *** *p* < 0.01.

**Figure 7 molecules-28-00290-f007:**
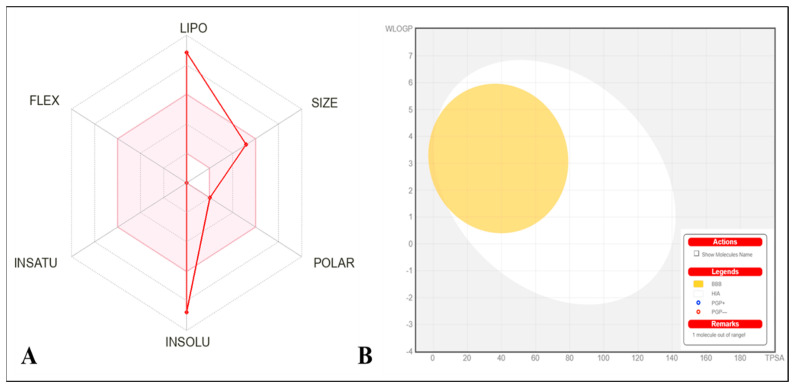
(**A**) Radar chart representing bioavailability pattern of EFD. The pink zone indicates the oral bioavailability physicochemical space, while the red line reflects the oral bioavailability parameters. (**B**) BOILED-Egg plot predicted for EFD using the Swiss ADME online web tool.

**Table 1 molecules-28-00290-t001:** Ligands’ binding affinity for target proteins. Residues of amino acids implicated in the interactions between the binding pocket and EFD.

Proteins	Doxorubicin	EFD	Amino Acid Residues Involved in the Binding Pocket Interactions
BAX	−9.0	−7.9	ILE(A:133), ARG(A:134), GLN(A:32). LYS(A:128), GLU(A:41), LEU(A:125), PRO(A:130), ALA(A:35), MET(A:38)
BCL-2	−8.5	−8.1	ARG(A:129), GLY(A:8), TYR(A:120), GLU(A:9), PHE(A:128), LEU(A:11), VAL(A:10), PRO(A:125)
NF-κB	−2.0	−1.9	ASP(B:129), PRO(B:127), LYS(B:128)
P53	−7.6	−8.5	THR(A:329), PHE(A:328), LEU(A:330), ASN(A:345), HOH(A:1004), HOH(A:1010), ARG(A:342), PHE(A:338), ILE(A:332),PHE (A:338)

Note: Binding energy (E-value) is represented as kcal/mol.

**Table 2 molecules-28-00290-t002:** Assessment of size, weight and ROW of prostate after EFD treatment.

Groups	Weight of Whole Epididymis (g)	Weight of Both Testes (g)	Weight of Prostate (g)	Final Body Weight (g)	ROW
Control	0.459 ± 0.007 ^bc^	2.25 ± 0.12 ^b^	0.385 ± 0.03 ^d^	213 ± 8.0 ^c^	0.180 ± 0.006 ^c^
Vehicle (10 % DMSO)	0.456 ± 0.021 ^b^	2.23 ± 0.18 ^b^	0.379 ± 0.02 ^d^	208 ± 11 ^b^	0.182 ± 0.008 ^c^
BPA (50 mg/kg)	0.334 ± 0.007 ^a^	1.68 ± 0.29 ^a^	0.478 ± 0.08 ^a^	174 ± 13.0 ^a^	0.274 ± 0.019 ^a^
EFD (10 mg/kg)	0.468 ± 0.056 ^c^	2.32 ± 0.18 ^c^	0.388 ± 0.06 ^d^	221 ± 9.0 ^d^	0.175 ± 0.008 ^d^
BPA + EFD (10 mg/kg)	0.462 ± 0.063 ^bc^	2.28 ± 0.11 ^bc^	0.404 ± 0.05 ^c^	221 ± 8.0 ^d^	0.182 ± 0.008 ^c^
BPA + EFD (05 mg/kg)	0.454 ± 0.043 ^b^	2.26 ± 0.17 ^b^	0.429 ± 0.04 ^b^	215 ± 10.0 ^c^	0.199 ± 0.010 ^b^

Note: ROW, relative organ weight, BPA, Bisphenol A. Data values represent Mean ± SD (*n* = 7). Means with dissimilar superscript (^a–d^) letters in the column are significantly (*p* < 0.05) different from one another.

**Table 3 molecules-28-00290-t003:** Effect of EFD on hematological parameters.

Groups	RBCs (×10^6^)/µL	WBCs (×10^3^)/µL	Platelets (×10^3^)/µL	Hb (g/dL)	ESR (mm/h)
Control	5.93 ± 0.26 ^d^	3.93 ± 0.32 ^c^	514.81 ± 4.28 ^e^	12.33 ± 0.79 ^de^	3.97 ± 0.25 ^d^
Vehicle (10 % DMSO)	6.08 ± 0.28 ^e^	3.81 ± 0.45 ^d^	499.81 ± 4.58 ^d^	12.45 ± 0.91 ^e^	4.02 ± 0.45 ^d^
BPA (50 mg/kg)	3.88 ± 0.25 ^a^	7.37 ± 0.35 ^a^	308.67 ± 2.51 ^a^	6.57 ± 0.12 ^d^	9.19 ± 0.61 ^a^
EFD (10 mg/kg)	5.83 ± 0.19 ^cd^	4.17 ± 0.11 ^bc^	489.4 ± 1.51 ^d^	12.13 ± 0.47 ^d^	4.13 ± 0.03 ^cd^
BPA + EFD (10 mg/kg)	5.51 ± 0.12 ^c^	4.32 ± 0.14 ^bc^	451.2 ± 1.71 ^c^	11.26 ± 0.71 ^c^	4.31 ± 0.072 ^c^
BPA + EFD (05 mg/kg)	5.12 ± 0.31 ^b^	4.85 ± 0.17 ^b^	405.1 ± 1.91 ^b^	10.12 ± 0.25 ^b^	5.12 ± 0.101 ^b^

Note: Results are presented as mean ± SD (*n* = 7). Means with different superscript (^a–e^) letters in the column are significantly (*p* < 0.05) different from one another.

**Table 4 molecules-28-00290-t004:** Appraisal of reforms to hormonal levels by EFD.

Groups	Testosterone (ng/mL)	FSH (mIU/mL)	LH (mIU/mL)	Estradiol (pg/mL)
Control	4.41 ± 0.12 ^d^	11.27 ± 0.52 ^e^	3.14 ± 0.23 ^d^	18.02 ± 0.94 ^e^
Vehicle (10 % DMSO)	4.33 ± 0.19 ^d^	10.93 ± 0.75 ^d^	3.29 ± 0.33 ^d^	19.14 ± 1.17 ^d^
BPA (50 mg/kg)	1.47 ± 0.16 ^a^	5.71 ± 0.31 ^a^	1.41 ± 0.14 ^a^	26.19 ± 2.16 ^a^
EFD (10 mg/kg)	4.23 ± 0.15 ^c^	10.59 ± 0.18 ^cd^	3.14 ± 0.09 ^d^	19.44 ± 0.41 ^d^
BPA + EFD (10 mg/kg)	4.02 ± 0.09 ^bc^	10.12 ± 0.27 ^c^	2.90 ± 0.13 ^c^	21.07 ± 1.17 ^c^
BPA + EFD (05 mg/kg)	3.81 ± 0.16 ^b^	8.62 ± 0.36 ^b^	2.49 ± 0.12 ^b^	22.82 ± 0.88 ^b^

Note: FSH, follicle stimulating hormone, LH, luteinizing hormone, BPA, bisphenol A. All the data are represented as Mean ± SD (*n* = 7), Means with different superscript letters (^a–e^) in a column specify significant difference at *p* < 0.05.

**Table 5 molecules-28-00290-t005:** Effect of EFD on biochemical levels.

Groups	CAT (U/min)	POD (U/min)	SOD (U/min)	GSH (μM/mg Protein)
Control	3.77 ± 0.12 ^d^	9.14 ± 0.63 ^d^	18.02 ± 1.54 ^e^	24.81 ± 3.14 ^f^
Vehicle (10 % DMSO)	3.83 ± 0.09 ^d^	9.29 ± 0.39 ^d^	18.14 ± 1.37 ^e^	23.68 ± 2.91 ^e^
BPA (50 mg/kg)	1.71 ± 0.07 ^a^	3.41 ± 0.14 ^a^	8.19 ± 1.56 ^a^	12.71 ± 2.84 ^a^
EFD (10 mg/kg)	3.59 ± 0.18 ^cd^	9.11 ± 0.29 ^d^	17.14 ± 1.91 ^d^	22.63 ± 3.19 ^c^
BPA + EFD (10 mg/kg)	3.27 ± 0.27 ^c^	7.99 ± 0.43 ^c^	16.67 ± 2.45 ^c^	21.67 ± 2.14 ^c^
BPA + EFD (05 mg/kg)	3.02 ± 0.21 ^b^	7.01 ± 0.21 ^b^	14.54 ± 1.58 ^b^	19.22 ± 2.43 ^b^

Note: All the data are represented as Mean ± SD (*n* = 7), Means with different superscript letters (^a–f^) in a column specify significant difference at (*p* < 0.05). CAT, catalase, POD, peroxidase, SOD, Superoxide dismutase and GSH, reduced glutathione.

**Table 6 molecules-28-00290-t006:** EFD’s predicted physicochemical and lipophilic profile.

Properties	Parameters	EFD
Physicochemical properties	MW (g/mol)	428.73
	Rotatable bonds	0
	HBA	1
	HBD	1
	Fraction Csp3	1.00
	TPSA	20.23
Lipophilicity Log *P_o/w_*	iLOGP	4.84
	XLOGP3	9.94
	MLOGP	7.07
	Consensus	7.30

(MW) Molecular weight, (HBA) H-bond acceptor, (HBD) H-bond donor, (TPSA) Topological polar surface area.

**Table 7 molecules-28-00290-t007:** Prediction of the ADMET profile of EFD.

Properties	Parameters	EFD
Absorption	Water Solubility	−6.49
	Caco permeability (cm/s)	1.389
	GI	100
	Log K_p_ (Skin permeation) cm/s	−2.716
	P-gp substrate	No
Distribution	BBB	0.72
	CNS permeation (Log PS)	−1.673
	V_D_ (human)	0.037
Metabolism	CYP1A2 inhibitor	No
	CYP2C19 inhibitor	No
	CYP2C9 inhibitor	No
	CYP2D6 inhibitor	No
	CYP3A4 inhibitor	No
Excretion	Total Clearance (log mL/min/kg)	0.023
	Renal OCT2 substrate	No
Toxicity	AMES toxicity	No
	hERG I inhibitor	No
	hERG II inhibitor	No
	Hepatotoxicity	No
	Skin Sensitization	No
	*T. Pyriformis* toxicity	0.354
	Minnow toxicity	−2.303

(GI) Gastrointestinal, (BBB) Blood-brain barrier, (V_D_) Volume of distribution, (AMES) Carcinogenic potential test, (hERG) the human Ether-à-go-go-Related Gene.

## Data Availability

The data used to support the findings of this study are available from the corresponding author upon request.

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
