# Peer review of "Ameliorative Effect of Structurally Divergent Oleanane Triterpenoid, 3-Epifriedelinol from Ipomoea batatas against BPA-Induced Gonadotoxicity by Targeting PARP and NF-κB Signaling in Rats"

_molecules, 2022, doi:10.3390/molecules28010290_

Round 1

Reviewer 1 Report

The manuscript is very interesting, well organized and has a rationale.

Perhaps a graphical abstract could provide an immediate result of the results obtained.

I suggest adding these manuscripts (see below) in the introduction to emphasize the importance of medicinal plants and their potential help for people's health.

Mechchate, H.; Ouedrhiri, W.; Es-safi, I.; Amaghnouje, A.; Jawhari, F.z.; Bousta, D. Optimization of a New Antihyperglycemic Formulation Using a Mixture of Linum usitatissimum L., Coriandrum sativum L., and Olea europaea var. sylvestris Flavonoids: A Mixture Design Approach. Biologics 2021, 1, 154-163.

Ferrarini EG, Paes RS, Baldasso GM, de Assis PM, Gouvêa MC, Cicco P, Raposo

NRB, Capasso R, Moreira ELG, Dutra RC. Broad-spectrum cannabis oil ameliorates

reserpine-induced fibromyalgia model in mice. Biomed Pharmacother. 2022 Aug

18;154:113552.

Do the authors have information about the safety?

Do the authors think that the microbiota can interfere? the authors can read these articles where the role of the microbiota with medicinal plants and also with various pathologies is evident

Dargahi, N.; Johnson, J.C.; Apostolopoulos, V. Immune Modulatory Effects of Probiotic Streptococcus thermophilus on Human Monocytes. Biologics 2021, 1, 396-415

Duygu AÄŸagündüz, Betül Kocaadam-Bozkurt, Osman Bozkurt, Heena Sharma, Renata Esposito, Fatih ÖzoÄŸul, Raffaele Capasso. Microbiota alteration and modulation in Alzheimer's disease by gerobiotics: The gut-health axis for a good mind. Biomedicine & Pharmacotherapy, 2022

Pag. 2 please write Ipomoea batatas in italics “Ipomoea batatas

why did the authors not do the EFD group (5 mg/kg) alone? With whom is the effect of BPA+EFD (5 mg/kg) compared?

In the Discussion, the Authors should highlight the possible clinical significance of their findings

Author Response

I am grateful that reviewer has carefully been through the manuscript and raised very technical points and some typographic mistakes. This revision will definitely help to enhance the quality of present study and Data.

Comment:

Perhaps a graphical abstract could provide an immediate result of the results obtained.

Response:

Graphical abstract has been added to the manuscript as per suggestion.

Comment:

I suggest adding these manuscripts (see below) in the introduction to emphasize the importance of medicinal plants and their potential help for people's health.

Response:

All suggested references has been added to the manuscript.

Comment:

Do the authors have information about the safety?

Response:

Thorough investigation for the toxicity profiling has been conducted for compound. Complete investigation via MTT assay on blood lymphocytes, cryopreserved hepatocytes and chondrocytes, Comet assay on blood lymphocytes and gel electrophoresis on extracted DNA has been conducted. Acute, sub-chronic and chronic toxicity assessment of the compound has been done to establish its safety profiling in vivo. Similarly toxicity of the compound has been assessed using in silico methods. A part of which has been added (ADMET) to the manuscript. But all set of multimode safety profiling data has been submitted to another journal as separate project.

Comment:

Do the authors think that the microbiota can interfere? the authors can read these articles where the role of the microbiota with medicinal plants and also with various pathologies is evident

Response:

I highly acknowledge the reviewer to introduce us with this wonderful microbiota concept. The use of microbiota along with potent phytochemicals especially in the form of supplements can surely change the therapeutic approach in the world full of diseases and disorders. Developing such therapeutic products can surely increase patient compliance and treat inflammatory disorders in cost effective way with no side effects. This wonderful concept has been added to the manuscript.

Comment:

Please write Ipomoea batatas in italics “Ipomoea batatas”

Response:

All typo mistakes have been rectified accordingly.

Comment

Why did the authors not do the EFD group (5 mg/kg) alone? With whom is the effect of BPA+EFD (5 mg/kg) compared

Response:

Dose selection in the current study was done following OECD guidelines. Firstly acute toxicity assessment was done using 10 mg/kg, 20 mg/kg, 50 mg/kg, 100 mg/kg and 250 mg/kg. Slight toxic symptoms like diarrhea, salivation and less food uptake was observed in group receiving 50 mg/kg. But the maximum toxic dose (MTD) was found as 100 mg/kg which killed one rat. It was then divided by 10 time to define high dose of 10 mg/kg and 50 mg/kg was divided by 10 time to select low dose (5 mg/kg). Multidose toxicity of the compound at dose of 10 mg/kg was also assessed throughout the study. So to assess minimum effective dose (MEC/MED), low and dose strategy was followed.

Comment

In the Discussion, the Authors should highlight the possible clinical significance of their findings.

Response:

Discussion and conclusion of the current study has been revised as per reviewer’s suggestion and clinical significance has been added.  

Reviewer 2 Report

The manuscript reports a good level of data. A series of in silico, in vitro, and in vivo experiments were conducted to detect the pharmacological activity of the triterpenoid isolated from the aerial part of Ipomoea batatas.

It should be indicated in an Abstract and among keywords that exactly aerial part of Ipomoea batatas was used to obtain the extract in this study.

line 4 of Abstract – maybe the word 'derived' replace with 'isolated' in '(EFD) derived from Ipomoea batatas'

page 2 and list of references: The Latin name of species should be written in italic type everywhere: Ipomoea batatas.

The Conclusion could be improved and contain information that the triterpene compound was isolated specifically from the aerial parts of plants and its name/discription can be given.

Why weren't used enough references published in the last 3 years? There is only 1 source for each of theіу years: 2022, 2021 and 2020. Such publications can be easily found in PubMed or Scopus databases. These additions would significantly strengthen the novelty of the chosen topic.

For instance: https://pubmed.ncbi.nlm.nih.gov/35458672/

In my opinion, there are too many self-citations in the list of used sources (for example, for the first author Muhammad Majid there are as many as they are: NN 9, 12,  15, 17, 19, 20, 21, 41, 42, 44)

Author Response

I really appreciate that reviewer has carefully been through the manuscript and raised very technical points as well as some typos. All suggested corrections has been carefully made to enhance the quality of manuscript.

Comment:

It should be indicated in an Abstract and among keywords that exactly aerial part of Ipomoea batatas was used to obtain the extract in this study.

Response:

Aerial part of I. batatas has been added to abstract and key words.

Comment:

line 4 of Abstract – maybe the word 'derived' replace with 'isolated' in '(EFD) derived from Ipomoea batatas.

Response:

Respective change has been done accordingly.

Comment:

In Figure 3 A and B, the lines should be colored to clear.

Response:

Respective changes have been made to the figure.

Comment:

page 2 and list of references: The Latin name of species should be written in italic type everywhere: Ipomoea batatas.

Response:

Thorough revision of the manuscript was done and made sure to remove all typos and rectify it as suggested.

Comment:

The Conclusion could be improved and contain information that the triterpene compound was isolated specifically from the aerial parts of plants and its name/discription can be given.

Response:

Conclusion of the study has been updated accordingly.

Comment:

Why weren't used enough references published in the last 3 years? There is only 1 source for each of theіу years: 2022, 2021 and 2020. Such publications can be easily found in PubMed or Scopus databases. These additions would significantly strengthen the novelty of the chosen topic.

Response:

Newest references has been added been added and self-citation has been minimized.

Reviewer 3 Report

This manuscript by Muhammad Majid et al. have investigated the anticancer activity and the ameliorative effect of 3-epifriedelinol from Ipomoea batatas against BPA-induced gonadotoxicity by targeting PARP and NF-κB signaling in rats. In order to investigate the therapeutic potential and anticancer activity to deal with prostate cancer (PCa) and gonadotoxicity, this study integrated a number of in vitro, in silico, and in vivo experiments. While this information, in general, is of interest and add to the current literature illustrating the role of 3-epifriedelinol in alleviating gonadotoxicity and the anti cancer potentail, there are several major critiques that have to be addressed by the authors and I have some concerns regarding the presentation, method design and interpretation of the data.

First, all abbreviations should be fully described in their first mention.

Generally, the abstract should be rewritten. It is not informative and does not reflect the aim of the research. The authors focused on the data of the in vitro experiment without clarifying the major findings of the in vivo experiment. In addition the conclusion not clear in view of the obtained results.

 The introduction should be revised or rewritten because the majority of the information didn't not focus on the tested compound 3-epifriedelinol and its role in the field of cancer and toxicology.

The study's objectives are unclear, so the authors should explicitly state the hypothesis and study objectives in detail.

To determine whether the tested substance (3-epifriedelinol) has a deleterious effect on normal cells, did the authors employ healthy normal cells for the toxicity test?

The authors should explain why only one type of cancerous cell (DU145) was used in the western blot study.

The authors must defend how they chose the 3-epifriedelinol dosages for the animal experiment.

The authors should supplement a figure about survival rate with time curve.

Please correct the manuscript for language and grammatical mistakes.

Author Response

Kindly find the attachment for a response.

Round 2

Reviewer 3 Report

The authors satisfactorily handled the majority of my concerns. Therefore, I strongly advise that this work be published in the journal molecules.